# Drug Resistance: An Incessant Fight against Evolutionary Strategies of Survival

**Francisco Ramos-Martín *** and **Nicola D'Amelio ***

Unité de Génie Enzymatique et Cellulaire UMR 7025 CNRS, Université de Picardie Jules Verne, 80039 Amiens, France

* Correspondence: francisco.ramos@u-picardie.fr (F.R.-M.); nicola.damelio@u-picardie.fr (N.D.); Tel.: +33-3-22-82-74-73 (F.R.-M. & N.D.)

**Abstract:** The inherent capacity of all organisms to evolve under selective pressure has created multidrug-resistant microbes and cells that are now threatening decades of progress in human health introduced by the advent of antibiotics. This resistance is developed by all pathogens, from bacteria to cancer cells, through fungi, parasites, or the seemingly simpler entities, i.e., viruses. In this review, we give an overview on this phenomenon, describing the mechanisms by which resistant organisms manage to evade the action of drugs. We also review existing therapies, including some of the most recent. This bibliographic review shows how, despite the encouraging progress that has been achieved in many areas, a permanent effort from scientists is necessary and will always be needed in order to compensate for the continuous microbial response intrinsically linked to the evolutionary process.

**Keywords:** bacteria; virus; parasites; cancer; fungi; multidrug resistance



## 1. Introduction

Drug resistance is a major public health problem that affects any illnesses caused by microorganisms or cancer cells and is a burden for those suffering from other diseases, who are more vulnerable to microbial colonization. Resistance arises in both prokaryotic and eukaryotic species and accounts for a significant number of deaths each year [1–3]. Resistance is inherent in evolutionary processes; therefore, the hunt for new antimicrobial agents and techniques is (and will always be) necessary. Pathogen mutability and reproduction rate are critical in the development of resistance mechanisms that are likely to be passed to other pathogenic organisms.

It is important to point out that resistance is not exclusively observed in bacteria and can be also found in other organisms such as fungi, viruses, or parasites. In the case of viruses, resistance can be developed to antiviral drugs that usually interfere with mechanisms of viral replication. If the treatment is not sufficiently effective, selective pressure may result in adaptation and resistance [4], a phenomenon exacerbated by the high rates of mutation characterizing many viruses. Resistance established by eukaryotic organisms (such as fungi and parasites) is more difficult to be dealt with because they share more traits with animal hosts than viruses or prokaryotes [5]. Finally, host cells undergoing malignant transformation can develop resistance against several drugs impairing their action [5,6].

In this review, we will mostly focus on bacterial and viral resistance, but we will also discuss other important pathogens such as fungi, parasites, and even cancer cells.

## 2. Bacterial Resistance

The persistent use of antibiotics, self-medication, and exposure to nosocomial infections has provoked the emergence of multidrug resistance bacteria (MDR) worldwide [7–10]. The term ESKAPE was adopted to refer to some of the most relevant pathogens displaying

the highest risk of mortality: *Enterococcus faecium*, *Staphylococcus aureus*, *Klebsiella pneumoniae*, *Acinetobacter baumannii*, *Pseudomonas aeruginosa*, and *Enterobacter* [9,10]. ESKAPE bacteria and extensively resistant *Mycobacterium tuberculosis* were listed by the World Health Organization (WHO) within the 20 bacteria against which new antimicrobials are urgently needed [11]. Three categories are specified with critical, high, and medium priority. Carbapenem-resistant *A. baumannii* and *P. aeruginosa* along with extended-spectrum β-lactamases (ESBLs) or carbapenem-resistant *K. pneumoniae* and *Enterobacter* spp. are listed in the critical-priority category of pathogens. In the high-priority group, we can find vancomycin-resistant *Enterococcus faecium* (VRE), methicillin- or vancomycin-resistant *Staphylococcus aureus* (MRSA and VRSA, respectively), clarithromycin-resistant *Helicobacter pylori*, fluoroquinolone-resistant *Campylobacter* spp., *Salmonella* spp., and fluoroquinolone- or third-generation cephalosporin-resistant *Neisseria gonorrhoeae*. Finally, *Streptococcus pneumoniae*, *Haemophilus influenzae*, and *Shigella* spp. are in the medium-priority category [11].

The mechanisms of resistance (detailed in Section 2.3) are broadly grouped into different categories: drug inactivation (commonly characterized by an irreversible cleavage catalyzed by an enzyme), modification of the antibiotic binding site, and reduced accumulation of the drug either due to reduced membrane permeability or by increased efflux of the drug [12]. The gene responsible for one resistance mechanism can be passed on to bacteria holding genes for other mechanisms, resulting in multidrug-resistant species, which are the source of our current health crisis. This is due to a mechanism known as "horizontal gene transfer", in which a bacterium receives resistant genes directly from the environment (transformation), from another bacterium in the form of plasmids (conjugation), or by phages (transduction). Horizontal gene transfer is also seen in cancer cells [13–16].

Other resistance mechanisms are present in bacteria aimed at evading both innate and cellular immune response (see Section 2.3).

## 2.1. Bacterial Targets and Main Class of Antibiotics

Strategies to overcome infections rely on targeting bacterial structures absent or significantly different in mammalian cells to prevent toxicity. For this reason, the bacterial wall, absent in mammalian cells, is one of the most important targets. Additionally, a large number of antibiotics have been developed targeting DNA replication, protein synthesis, or bacterial metabolism (Figure 1).

The inhibition of cell-wall synthesis can be achieved by drugs (β-lactams) binding to the bacterial transpeptidases (also called penicillin-binding proteins, PBPs), thus impeding cross-linkage between the peptidoglycan polymer chains. Alternatively, as in the case of vancomycin, the linkage is inhibited by the binding of the antibiotic to the peptidic portion of the chain. The evolution of bacterial β-lactamases (enzymes able to disrupt the chemical structure of β-lactams) directed the evolution of the first naturally derived penicillins into different classes of β-lactam antibiotics called cephalosporins, cephamycins, monobactams, and carbapenems [17]. Carbapenems are considered as the last line of defense in complex infections. Whereas resistance mechanisms in Gram-negative bacteria are often characterized by the expression of β-lactamases (both chromosomal or acquired by plasmids or integrons), mutations of penicillin-binding proteins are more frequently found in Gram-positive bacteria [18]. In particular, three gene types are of particular concern because of their carbapenemase activity: class A KPC and GES, class D OXA, and class B metallo-β-lactamases (MBLs) [19]. In European hospitals, common cases of resistance are found with *P. aeruginosa* expressing VIM-2 and overexpressing AmpC, or Enterobacteriaceae expressing OXA-48 or KPC-2. In other cases, the overexpression of non-carbapenemase AmpC causes the sequestration of the antibiotic without its degradation [20].

Another target for antibiotics is the bacterial membrane, whose composition in phospholipids (rich in phosphatidylglycerol (PG) and LPS in Gram-negative bacteria, and occasionally cardiolipin (CL) [21]) significantly differs from the mammalian one. Antimicrobial peptides and polymyxins are able to selectively interact with bacterial membranes and create pores leading to bacterial death [22]. Polymyxin B and polymyxin E (colistin) are

the last resort for treating infections caused by multidrug-resistant Gram-negative bacteria. They are able to disrupt the outer membrane of Gram-negative bacteria by displacing $Ca^{2+}$ and $Mg^{2+}$ ions which bridge lipopolysaccharide (LPS) molecules [23]. SPR741 [24] is a potent alternative to polymyxin B and does not show the severe nephrotoxic effects of the latter. Daptomycin is a lipopeptide active only on Gram-positive bacteria. All these compounds can also potentiate the effect of different antibiotics by granting them the access to their intracellular targets [25]. The reduced permeability of the membrane can also be partially addressed by the development of siderophore antibiotics, compounds able to bind iron and pass the outer membrane using the iron uptake system of Gram-negative bacteria [26].

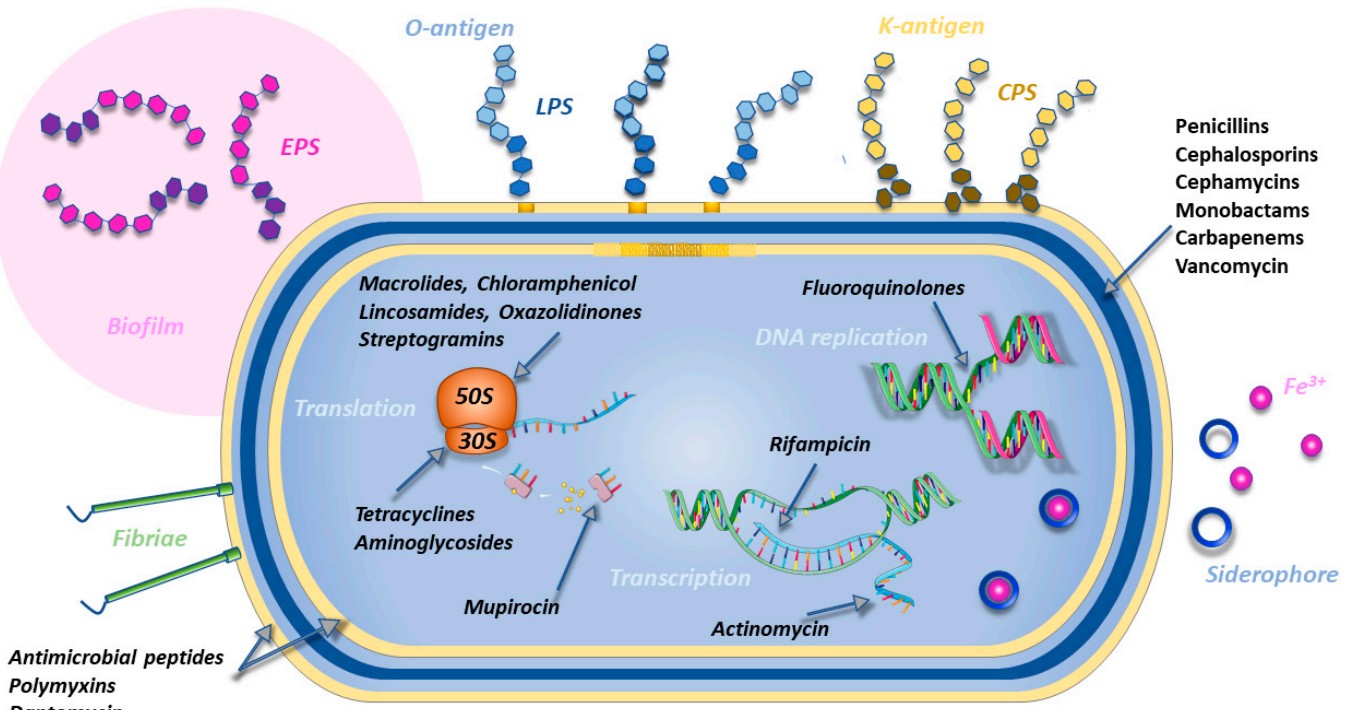

**Figure 1.** Classes of antibiotics acting at the level of different bacterial processes (cell-wall synthesis, membrane assembly, DNA replication, transcription, and translation) and virulence factors correlated with resistance (LPS O-antigens, CPS K-antigens, biofilms, fimbriae, and siderophores). Own work using a few icons by Servier Medical Art.

Due to the different structure of the bacterial machinery, antibiotics have been developed interfering with bacterial DNA and protein synthesis. The synthesis of DNA can be inhibited by fluoroquinolones which target bacterial DNA gyrase and topoisomerase IV, two enzymes important for DNA replication [27]. The transcription to RNA can be blocked by the inhibition of bacterial DNA-dependent RNA polymerase (rifampicin) or the suppression of RNA elongation due to drugs binding to the transcription initiation complex (actinomycin) [28,29]. Another important target for the inhibition bacterial protein synthesis is the ribosome. Antibiotics can bind to the 30S (tetracyclines, aminoglycosides) or to the 50S subunits (macrolides, lincosamides such as clindamycin, oxazolidinones such as linezolid or tedizolid, and streptogramins) [30–33]. Tetracyclines are bacteriostatics that prevent the attack of the first transfer RNA (which in bacteria often codes for methionine) while aminoglycosides such as streptomycin, neomycin, or gentamicin bind to the A-site on the 16S ribosomal RNA of the 30S subunit, leading to the misreading of the genetic code and inhibition of translocation. Macrolides such as erythromycin and chloramphenicol must be used under medical supervision, due to their inhibition of bone marrow function. Resistance to macrolides, lincosamides, and streptogramins can occur via the methylation of their receptor binding site on the ribosomes [32]. Protein synthesis can be blocked by

inhibiting other targets. For example, rifampin binds to RNA polymerase, while mupirocin inhibits isoleucine tRNA synthetase [32].

Finally, sulfonamides are drugs that act on bacterial metabolism by interfering with the production of folic acid which is the precursor of purines, pyrimidines, and amino acids [34].

### 2.2. Resistant Health-Threatening Bacteria

#### 2.2.1. *Klebsiella pneumoniae*

*K. pneumoniae* is a Gram-negative and rod-shaped bacterium, commonly found in the human nose and gastrointestinal tract. Resistant strains of *K. pneumoniae* represent a threat to human health because of their incidence as nosocomial infections, affecting patients after surgeries or transplants [2,35–37]. *K. pneumoniae*, which has recently gained resistance to carbapenems and is able to form biofilms, disseminates from its common location and creates severe infections [36,38,39]

*K. pneumoniae* MDR strains often owe their properties to resistance plasmids and transferable genetic elements [40], which has resulted in the emergence of extremely drug-resistant strains with an accumulation of antibiotic resistance genes [41]. The main virulence factors identified are pili, capsule, LPS, and iron carriers. *K. pneumoniae* adheres to epithelial immune cells by adhesins of type 1 and type 3 pili. Strains containing the *RmpA* plasmid, which regulates the synthesis of capsular polysaccharides protecting from immune responses, produce purulent tissue infections as liver abscess [41]. The modification of LPS lipids impairs recognition by the immune cells and blocks the consequent inflammatory response leading to the clearance of the bacteria. Four iron carriers are used by *K. pneumoniae*: enterobactin, yersiniabactin, salmochelin, and aerobactin. They exist in typical and highly virulent strains, but the latter produce larger amounts of active iron-absorbing molecules [41,42]. New virulence factors have been described for highly virulent strains. For example, some produce toxic colibactin, which damages host DNA [41].

Other resistance mechanisms involve drug-modifying enzymes able to adenylate, acetylate, or phosphorylate their targets. Resistant strains are able to use them to develop resistance against beta-lactam antibiotics, polymyxins, or quinolones [41,43].

#### 2.2.2. *Pseudomonas aeruginosa*

*P. aeruginosa* is also a Gram-negative and rod-shaped bacterium, and a common opportunistic and nosocomial pathogen that causes mortality in cystic fibrosis patients. Some phenotypic variants are highly resistant and possess an enhanced ability to form biofilms involved in ventilator-associated pneumonia, bacterial keratitis, otitis externa, and infections of peritoneal dialysis catheters, burn wounds, and the urinary tract [44]. Interestingly, the membrane of this bacterium changes in composition when involved in biofilms, showing a decrease in branched chain phospholipids and the creation of longer chains of phosphatidylethanolamine that results in lower fluidity and higher stability. Reducing membrane fluidity seems to be a strategy to conserve energy, as bacteria in biofilms are less metabolically active [44]. The role of membrane composition and pathogenesis was recently reviewed by us [21].

MDR strains of *P. aeruginosa* are able to bypass the action of most antibiotics because of their high levels of intrinsic and acquired resistance mechanisms [45,46]. Intrinsic mechanisms include lower outer membrane permeability, the use of efflux pumps to expel antibiotics, and the expression of enzymes able to inactivate antibiotics [46]. Some strains have acquired resistance thanks to horizontal gene transfer processes or mutations. Finally, the formation of biofilms is an adaptive mechanism of resistance causing prolonged and recurrent infections. Persistent cells in a dormant state inactivate the synthesis of the antibiotic targets, thus remaining viable and ready to repopulate biofilms [45,47,48].

A total of 26 porins regulate the penetration of antibiotics through the bacterial membrane of *P. aeruginosa* or modulate its adhesion to animal cells. Their expression plays a major role in resistance [46]. For example, the reduced expression of OprD impairs the

entrance of beta-lactams and carbapenems, while the expression of OprF porin is required for adhesion to target cells and plays a major role in lung infections [46,49].

Efflux pumps are classified into five families: adenosine-triphosphate-binding cassette (ABC), the major facilitator superfamily (MFS), the multidrug and toxic compounds extrusion family (MATE), the resistance-nodulation-cell-division family (RND), and the small MDR family (SMR) [46]. RND pumps are the most important in *P. aeruginosa*; they are powered by proton motor force and can export molecules from the periplasm and the cytosol into the environment [46].

LPS molecules in the outer membrane, which are targeted by positively charged antimicrobials for their negative charge, can be modified through the attachment of 4-amino-4-deoxy-L-arabinose to the phosphate unit of its lipid A. Such a modification decreases the net negative charge of the bacterial membrane leading to a decrease in the affinity for positively charged antimicrobials and to an adaptation to a low concentration of $Mg^{2+}$, which normally stabilizes the outer membrane by bridging LPS molecules [46,50].

### 2.2.3. *Acinetobacter baumannii*

*A. baumannii* is a Gram-negative, almost round, nosocomial pathogen that causes ventilator-associated and bloodstream infections in critically ill patients. Some of the clinical manifestations are pneumonia, meningitis, and, among others, infections of skin soft tissues or the urinary tract [51]. Its plastic genome, which is able to mutate and face adverse conditions, generates MDR strains which have acquired methyltransferases, beta-lactamases, and high carbapenem, aminoglycoside, and colistin resistance [52].

*A. baumannii* is capable of surviving in dry environments (desiccation resistance). Because of this, it earned the name of Iraqibacter throughout the Afghanistan and Iraq conflicts [53]. In order to survive during periods of desiccation, this bacterium produces capsular polysaccharides that function as a glycan shield encompassing the entire cell protecting it from external threats, including complement-mediated death. In addition, its RecA enzyme (required for homologous recombination and repair) helps to prevent DNA lesions caused by desiccation. *A. baumannii* is also able to upregulate detoxifying proteins against reactive oxygen species (ROS), making some *Acinetobacter* spp. among the most hydrogen-peroxide-tolerant bacteria. Finally, it has been shown to possess pumps able to eject disinfectants like chlorhexidine [54] but also a micronutrient acquisition system, allowing the entry of iron-chelating molecules (siderophores) that could be exploited for drug delivery [54].

*A. baumannii* is able to form biofilms during skin and soft-tissue infections but also over most abiotic surfaces, such as health-care-associated equipment. Its hypermotility has been associated with increased virulence, as in the case of *Pseudomonas aeruginosa* [54].

LPS, which is the primary component of the outer leaflet of the outer membrane, has been suggested as an important pharmacological target [54,55]. Bacterial carbohydrates and glycans provide an interface between pathogens and their environment and are required for bacterial viability. Furthermore, *A. baumannii* can modify its lipid A (a part of LPS) to develop antibiotic resistance. For example, PE or galactosamine can be added to lipid A to develop resistance to colistin. Thin-layer chromatography analyses have shown that PE is an important component of its lipid extract together with cardiolipin (CL) and monolysocardiolipin, that are in surprisingly high ratios in this bacteria. CL and monolysocardiolipin might also be involved in lipid A remodeling, affecting the virulence of *A. baumannii* [56].

### 2.2.4. *Staphylococcus aureus*

*Staphylococcus aureus* is a Gram-positive bacterium whose antibiotic-resistant strains represent nearly half of all fatalities caused by resistant bacterial pathogens. Methicillin-resistant *Staphylococcus aureus* (MRSA) also evades non-beta-lactam antibiotics, such as lincosamides, macrolides, aminoglycosides, quinolones, and multiple antibiotic combinations [57]. Their biofilms and toxins can evade the host immune system leading to recurring

infections which are most frequently isolated in the skin and wounds [58,59]. In addition, *S. aureus* can invade host cells and persist and proliferate intracellularly, resulting in the formation of small-colony variants (SCVs). SCVs can be induced by the low pH of some intracellular compartments or found in inflamed tissues. They have reduced metabolic activity and are more tolerant to antibiotic therapy [58]. Some of these variants can infect neutrophils [60], which are the first line of defense against invading bacteria. *S. aureus* can even invade bones, leading to osteomyelitis, infecting osteocytes, osteoblasts, and osteoclasts intracellularly [61]. It can also attack cartilage and infect chondrocytes, synovial fibroblasts, and even immune cells of the synovial fluid [61]. *S. aureus* produces several virulence factors, such as the phenol-soluble modulins (PSMs) family of peptides. Its more cytotoxic members are peptides with roughly 20 residues that have amyloidogenic properties, antibacterial activity against other competing bacteria, and promote biofilm stability [62].

In this scenario, novel agents against intracellular antibiotic-tolerant *S. aureus* are urgently needed. Antimicrobial peptides (AMPs) have been used successfully to treat some of these infections [57], despite the fact that some strains have established resistance mechanisms [63] mostly involving lipid shedding [59,64]. Alternatively, while AMPs can also be combined with antibiotics for a synergic action [64,65], cell-penetrating peptides (CPPs) can be used to treat *S. aureus* infections intracellularly [58,66–71].

### 2.2.5. *Neisseria gonorrhoeae*

*N. gonorrhoeae* is a Gram-negative bacterium that causes the sexually transmitted disease gonorrhea [72]. According to WHO estimates, there were 82.4 million people newly infected with gonorrhea in 2020, an increase from the estimated 78 million new cases in 2012. The use of antibiotics has led to the development of resistance to all first-line drugs for its treatment. *N. gonorrhoeae* is an obligate human pathogen whose evolution is influenced by its infection niches. Phylogeny suggests that the lineage circulating primarily in heterosexuals has diverged in men that have sex with men. As a consequence, different resistance mechanisms have been observed in each lineage [73,74]. While vaccines should be theoretically efficacious, there has been little success in generating immune protection for this microorganism [72]. A new drug against gonorrhea should be able to tackle these resistance mechanisms and also be able to exert its action in multiple different tissues [72].

Genome content is not fixed in most bacteria; it is instead composed by a flexible gene pool named pangenome [75]. This can be divided into the core genome (with genes present in all isolates of a species) and the accessory genome (with genes present only in a subset of the population providing genetic adaptability). However, in the case of *N. gonorrhoeae*, the accessory genome content is largely unknown [76].

One of the main resistance mechanisms involves the four penicillin-binding proteins (PBPs) present in *N. gonorrhoeae*. PBP1 and PBP2 play a key role in beta-lactam resistance, while PBP3 and PBP4 have a lower molecular weight and show carboxypeptidase and endopeptidase activities [77,78]. Some strains have gained resistance to ceftriaxone and cefixime through mutations in the penA gene, especially in the domain encoding the PBP2 transpeptidase. However, other factors must contribute as different strains with these mutations show different vulnerability to both drugs [77].

Macrolide resistance is due to mutations affecting their binding to rRNA. At the same time, cumulative mutations in the DNA gyrase impair the action of many fluoro-quinolones [77,78] and ribosomal protection by the TetM, a protein encoded by conjugative plasmids that promotes resistance to tetracyclines [77].

### 2.2.6. *Mycobacterium tuberculosis*

*M. tuberculosis* infects almost a third of the world's population with almost two million deaths per year; it was the second deadliest infectious agent in 2021 after SARS-CoV2 [79,80]. Antibiotic development has considerably reduced mortality rates [81]. Most antibiotics can target the bacterial enzymes implicated in the synthesis of the bacterial wall, inhibit

bacterial RNA polymerase, or impair the action of some of their ribosomal proteins. In most of the cases, the treatment is highly effective, but drug resistance may arise due to the patient's poor adherence to the therapy, drug efflux, impermeability of the cell envelope [80,82], or the complexity of granulomas induced by the disease [79]. Granulomas are necrotic areas formed in the lungs which allow a long-term residence of the dormant pathogen in the host [83] and its dissemination in the population [79,83]. In recent years, strains resistant to rifampicin and isoniazid have appeared, with some also resistant to any fluoroquinolone and to at least one of the second-line antibiotics (kanamycin, capreomycin, and amikacin). The only way to restrain the spreading of these totally drug-resistant *M. tuberculosis* strains is currently early detection [84].

Rifampicin resistance is often produced by mutations in the "hot-spot" region of the *rpoB* gene, also called the rifampicin-resistance-determining region [85,86]. Isoniazid, together with rifampicin, is an essential antibiotic for the treatment, although it only acts on replicating bacteria. Isoniazid acts by inhibiting the synthesis of mycolic acids by the NADH-dependent enoyl–acyl carrier protein reductase. The main resistance mechanisms are linked to mutation in the *katG* and *inhA* genes, affecting the affinity for the isoniazid–NAD adduct [82,84].

*M. tuberculosis* presents resistance also to second-line antibiotics. For example, resistance to fluoroquinolones can arise from chromosomal mutation in the quinolone-resistance-determining regions of the gene coding for the two subunits of type II topoisomerase. Mutations at the level of 16S rRNA also promote resistance against kanamycin and amikacin, which act by inhibiting protein synthesis [80,84].

### 2.3. Bacterial Resistance Mechanisms

With the advent of the antibiotic era, several lineages with different resistance mechanisms have evolved (Figure 2). Initially, the broad use of penicillin resulted in an increase in strains carrying a beta-lactamase plasmid. This phenomenon has subsequently occurred for other antibiotics including tetracyclines, macrolides, quinolones, and cephalosporins. Resistance mechanisms can be grouped in different classes, including target modification, antibiotic inactivation, permeability reduction, drug extrusion by efflux pumps, ribosomal protection, and biofilm formation [87].

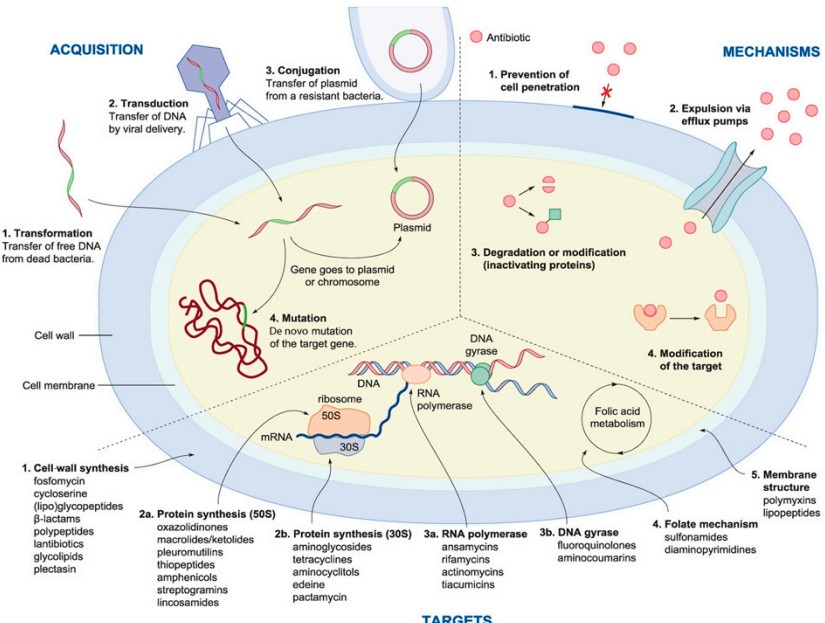

**Figure 2.** Multiple strategies that microbes use to develop resistance to drugs (**right**). The image also shows the mechanism for the acquisition of resistant genes (**left**) and (**bottom**) the main bacterial drug targets (from [87]).

### 2.3.1. Target Modification

Changing the interaction site of the antibiotic in the target is an essential mechanism of bacterial resistance. The modification can be achieved by random point mutations that have a minimal impact on bacterial cell homeostasis [52,87]. Common targets are proteins involved in cell-wall synthesis or the ribosome due to the fact that these structures are absent (cell wall) or profoundly different (ribosome) in humans. New targets are desperately needed to overcome drug resistance, possibly at the genetic level [87].

Another way to modify the target site is to alter its charge. For example, cationic AMPs (and other compounds) have affinity for negatively charged membranes. *Staphylococcus aureus* and several other Gram-positive bacteria are able to add positively charged groups (as protonated D-alanyl) to their cell wall, partially neutralizing the negative charge of the bacterial outer surface containing lipoteichoic acid (LTA). Alternatively, they are able to add positively charged lysine residues to the PG of their membranes [21], thus inhibiting AMP binding [63,65]. The use of different AMPs in combination can aid in overcoming this resistance mechanism [65].

### 2.3.2. Antibiotic Inactivation: The Case of β-Lactamases

β-lactamases are enzymes able to hydrolyze chemical compounds with a β-lactam ring, thus inactivating β-lactam antibiotics. They are ancestral enzymes dating back to about two billion years ago, while β-lactamases encoding plasmids appeared millions of years ago [88]. β-lactam antibiotics inactivate their target by acetylating a serine residue of penicillin-binding proteins (PBPs), which are transpeptidases necessary for bacterial cell-wall synthesis. In Gram-positive bacteria, the primary mechanism of β-lactam resistance is the alteration of the PBP affinity for these antibiotics while maintaining physiological functions. In Gram-negative bacteria, β-lactamase synthesis is the main mechanism of resistance to β-lactam antibiotics. The prevalence of β-lactamase synthesis in Gram-negative bacteria has been favored by the occurrence of transferable plasmids encoding a wide range of enzymes involved in the spread of β-lactam resistance [52,87].

### 2.3.3. Membrane Permeability Reduction

The permeability barrier provided by the outer membrane of Gram-negative bacteria typically leads to more resistant strains compared to Gram-positive bacteria. As this barrier prevents the antibiotics from gaining access to their targets inside the bacterial cell, the development of antibiotics to treat infections caused by Gram-negative bacteria is particularly challenging [87]. Bacteria also limit pore density by reducing porine expression [52,87]. AMPs can be used to increase the permeability of the outer membrane and allow antibiotics to reach their target inside the bacterial cell [87], although resistant strains can counteract their effect by changing membrane fluidity due to changes in phospholipid composition or overexpressing efflux pumps [64].

### 2.3.4. Phospholipid Shedding

Some bacteria, like some *S. aureus* resistant strains, have developed biochemical routes for the release of phospholipids to hijack the action of phospholipid-binding antibiotics, such as daptomycin or several AMPs, thus reducing their efficacy [63,64,89].

### 2.3.5. Drug Extrusion by Efflux Pumps

Efflux systems are membrane-associated protein complexes able to extrude antibiotics, effectively reducing their intracellular concentration [87]. They are predominantly found in Gram-negative bacteria, although they have been described also in Gram-positive bacteria. Efflux systems are usually found in mobile genetic elements (MGEs) (transposons, integrons, plasmids) which can be acquired from other organisms, thus facilitating their spread [52,87].



### 2.3.6. Ribosomal Protection

Ribosomal protection proteins are cytoplasmic proteins that reduce the susceptibility of the ribosome to antibiotic activity. They are found in both Gram-positive and Gram-negative bacteria, but they are more common in Gram-positive organisms. One example are the proteins used to prevent tetracycline action. These proteins cause allosteric disruption of the primary tetracycline binding site, leading to the release of bound tetracycline molecules. The ribosome is then able to return to its functional conformation and resume protein synthesis [87].

### 2.3.7. Evasion from Immune Surveillance

Bacteria causing intracellular infection exploit host protection (typically used by viruses) to hide from the immune machinery, although infected cells often expose PS on their surface as an eat-me signal [90].

Lipopolysaccharide (LPS) molecules at the surface of Gram-negative bacteria (Figure 1) can change the structure of their outermost part (O-antigen). For example, nine *K. pneumoniae* O-antigens were identified [91]. Such structures can inhibit the action of the complement by binding some of its proteins or mask the LPS innermost lipid A, which is highly immunogenic, and prevent recognition by immune cells. The chains of LPS can also protect the bacterial membrane from the action of antimicrobial peptides [92] and the components of the serum complement, especially when the modified chains maintain their full length (smooth phenotype) [93]. Bacteria can also significantly thicken their capsule by increasing the amount of capsule polysaccharide (CPS). The structure of such polymers can change in terms of the order and units of saccharide, exposing new moieties at the bacterial surface (K-antigens) which can either be immunogenic or avoid recognition by immune cells (thus becoming resistant). Most importantly, the capsule can prevent opsonophagocytosis [94] and hijack the serum complement system and antimicrobial peptides. It should be noted that most ESKAPE bacteria are encapsulated and capable of forming biofilms, which constitute another way to evade immune surveillance.

### 2.3.8. Biofilm Formation

When polysaccharide structures are secreted outside the membrane, they contribute to the formation of an extracellular polymeric substance (EPS), an important component of biofilms also containing proteins, secreted nucleic acids, humic substances, and metal ions (Figure 3) [95]. Biofilms can inactivate both the immune response and the effect of antibiotics. They not only shield susceptible cells but also specialized dormant cells that are relatively tolerant to antibiotics and cause recalcitrant infections [96]. Bacterial biofilms are formed by unicellular organisms in communities attached to a solid surface and encased in the EPS matrix. They can be composed of single or multiple bacterial species gathered by quorum sensing and intercellular communication. Their formation is essential for the activation of virulence and antibiotic resistance factors [97–99].

Bacteria in biofilms have more resilience than planktonic cells. Biofilms may grow on numerous medical devices, including catheters and contact lenses, and are linked to 65% of nosocomial infections [36]. The production of the glycocalyx, an EPS matrix, prevents the access of antibiotics to the bacterial cells. When bacteria go from exponential to slow or no growth, or when they are stressed, they boost their antibiotic resistance mechanisms [54,97,100]. Interestingly, bacteria in biofilms tend to be under stress and grow slowly because they experience nutrient limitations. Not all biofilms are involved in pathogenic or undesirable processes. For example, biofilms produced by our commensal microbiota have protecting roles in our gut [101]. In other cases, they are useful in industry for the production of biofuel because of the cellulolytic action of organisms such as *Clostridium thermocellum* [102].

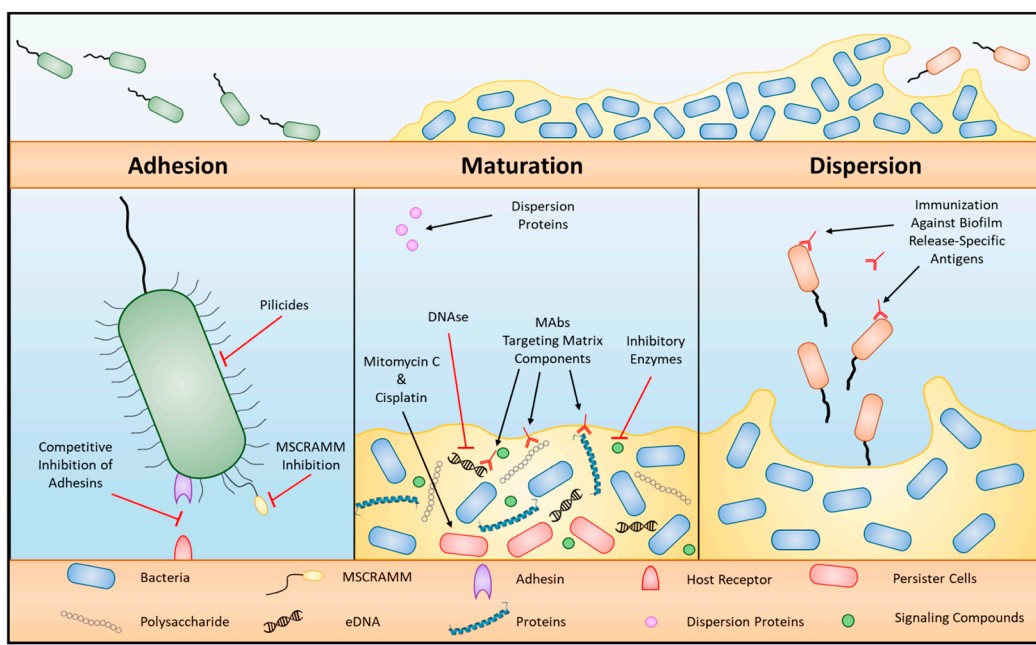

**Figure 3.** Biofilm formation and targets for therapy. Three general phases of biofilm formation: adhesion, maturation, and dispersion, with therapeutic targets per phase (from [103]).

### 2.3.9. Other Phenomena Correlated with Resistance

The correlation of resistance with the expression of fimbriae [104] or the presence of siderophores (small secreted molecules binding iron and re-importing it intracellularly) has also been observed [105]. Fimbriae are proteinaceous organelles [106] mediating the adherence with eukaryotic cells and/or macrophages, and they contribute to bacterial aggregation and the formation of biofilms [107,108].

### 2.4. Strategies to Overcome Drug Resistance in Bacteria

The most common strategy to treat bacterial infections involves the use of antibiotics alone or in combination [87]. Every year the number of antibiotics effective against ES-KAPE is declining, and it is then imperative to find alternative ways to treat their infections. Some of these approaches use antibiotics in combination with other antibiotics or adjuvants, bacteriophage therapy, novel drug delivery systems, photodynamic therapy, or the administration of AMPs, antibodies, or silver nanoparticles [109].

### 2.4.1. Combinations of Antibiotics

Combinations of antibiotics have been used to reduce resistance. While adjuvants have a synergistic effect, the use of multiple antibiotics increases the spectrum coverage [110]. Adjuvants are molecules with little or no antimicrobial activity but able to inhibit resistance mechanisms, blocking efflux pumps or attacking biofilms [111,112]. The most popular adjuvants are beta-lactamase inhibitors. The use of antibiotic hybrids [113], which are synthetic constructs of two or more pharmacophores from antimicrobial agents, holds the advantage of combined therapies while reducing the chances of resistance and overcoming the problem of non-complementarity in pharmacodynamic profiles of the individual antibiotics [10,113,114]. One example of such hybrids is the combination of beta-lactam antibiotics with a beta-lactamase inhibitor as in the case of clavulanic acid acting as a beta-lactamase inhibitor for beta-lactam amoxicillin [113].

### 2.4.2. Bacteriophage Therapy

Lytic phages against ESKAPE pathogens have been isolated and are easily available therapeutic agents [115]. They are viral parasites able to infect bacteria by injecting their genetic material into the host and replicating using the host cellular machinery. Genetically

modified phages could be used to increase the antibiotic susceptibility of resistant strains. Bacteriophages do not have adverse effects on the patient's microbiome and have a high degree of selectivity and specificity for pathogens [52]. Other advantages include low dosages for treatment and rapid proliferation. The high specificity of phages is both an advantage and a limitation. Their high specificity makes them prone to become inactive due to the modification of the target by the microorganisms. On the other hand, as they co-evolve alongside their host, they are capable of regaining their activity over newly evolved resistant bacteria [116,117]. Even more, their use in combination with other antimicrobials can bypass this issue while still being more specific than the other components of the "cocktail" alone [118].

Disadvantages include their poor stability, complex administration, and potential safety issues [10]. Genomic characterization of phages is important to predict their safety in therapeutic applications. Another disadvantage is that phages can promote the exchange of antibiotic resistance genes, thus enhancing horizontal gene transfection in bacteria [119]. They can also induce bacteria to acquire new resistance mechanisms due to the selective competition induced by the co-evolution with phages [52].

Phage-derived enzymes having the capacity to degrade peptidoglycan (the main component of the bacterial cell wall) are often called "enzybiotics". Their use is promising in the fight against antibacterial resistance.

Endolysins are the most studied peptidoglycan-degrading enzymes [120,121]. They are synthesized in the cytoplasm of the infected bacteria, and they usually require the action of holin proteins. Holins are phage-produced small hydrophobic proteins that oligomerize in the bacterial cytoplasmic membrane to form lethal holes allowing translocation of endolysin to the cell wall [121]. Endolysins quickly degrade the peptidoglycan polymers of the host, leading to osmotic lysis and the release of new phages [120,121].

Virion-associated lysins and endolysins have multiple enzymatic activities to degrade peptidoglycan. Peptidoglycan polymer is composed of N-acetylmuramic acid and N-acetylglucosamine residues linked by β-1,4 glycosidic bonds. Interconnected stem peptides cross-link the adjacent glycan strands, each of them linked to N-acetylmuramic acid residues thanks to an amide bond involving the first amino acid of the peptide stem. Depending on the bonds that the lysins and endolysins cleave, they can exert different activities: lysozyme, transglycosylase, glucosaminidase, amidase, or endopeptidase [120,121]. Their specificity and host range depend on the phage species. For example, lysins from Gram-positive and Gram-negative phages act on Gram-positive and Gram-negative bacteria, respectively, but they can be so specific to recognize a single serotype [122].

Endolysins can have a globular or a modular design. The modular ones have lytic enzymes constituting a single enzymatic catalytic domain responsible for cleaving a specific peptidoglycan bond. In the modular structure, one or two N-terminal enzymatic catalytic domains are connected by a linker to cell binding domains that recognize the specific epitopes on the cell wall. This modular form is more often found in phages targeting Gram-positive and mycobacteria [120,123]. They have emerged as novel antimicrobials and are effective against several MDR strains (such as *S. aureus*, *E. faecium*, among others). They can be used in combination with antibiotics and other drugs [120,123], and they can displace divalent cations ($Ca^{2+}$ and $Mg^{2+}$), contributing to membrane instability [123]. They can also be combined with cell-penetrating peptides to target intracellular infections efficiently [58].

Depolymerases are another class of phage enzymes that degrade polysaccharide molecules such as LPS, the bacterial capsule, or the biofilm matrix [122]. They are classified according to their mode of action into hydrolases and lyases, cleaving a glycosidic bond by trans-β-elimination. Endolysins and depolymerases have been successfully used against biofilms and capsules of several microorganisms such as *K. pneumoniae*, *A. baumannii*, *P. aeruginosa*, and others [124–129]. Resistance against depolymerases can develop due to modifications in the LPS molecule or the bacterial capsule. On the other hand, resistance is

a rare event for phage lysins, as they are able to bind and cleave highly conserved targets in the cell wall [130].

### 2.4.3. Antimicrobial Peptide Therapy

There exist thousands of AMPs reported to show in vitro and in vivo antimicrobial and antibiofilm activities. Besides being antimicrobial, they can be used to grant access inside bacteria to other antibiotics, overcoming the reduced membrane permeability observed in resistant strains and acting as enhancers. They can also be used as novel drug delivery systems, besides those commented upon further in Section 5. For example, tobramycin can be conjugated to a cell-penetrating peptide to allow its internalization in bacteria [131,132].

Because AMPs do not interact with specific molecules in pathogens (but more in general with the bacterial membrane), resistance to them evolves at a relatively modest rate [133]. Furthermore, their rapid pharmacodynamics and maximal killing rate is substantially higher than that of antibiotics, thus reducing the likelihood of mutations in target species [133,134].

### 2.4.4. CRISPR

One of the most attractive recent strategies to combat bacterial resistance is the use of the clustered regularly interspaced short palindromic repeat (CRISPR) system (Figure 4). CRISPR/Cas is an immune defense system found in bacteria able to recognize and degrade foreign nucleic acids (e.g., acquired from phages) through associated caspases. The CRISPR/Cas system can be engineered and used to cut vital genes of a specific bacterium resistant strain, with high specificity. This is the case of targeted-antibacterial-plasmids (TAPs) which deliver CRISPR/Cas systems to target vital genes of pathogenic bacteria by means of bacterial conjugation [135].

The full process requires three stages: acquisition, crRNA biogenesis and interferencee.

The acquisition stage involves the insertion of 26–72 pair long foreign sequences (called a protospacer and derived from phages or MGEs such as plasmids or transposons) among repetitive loci of the host chromosome called crRNA. Discrimination between self and non-self is accomplished by sequences from the foreign nucleic acid called the protospacer adjacent motif (PAM). Direct target recognition is achieved only by identifying these sequence motifs not stored in CRISPR loci [52,136].

Once the foreign DNA has been acquired, the expression consists of transcribing the repetitive and foreign sequences into a single RNA transcript. RNA sequences partially complementary to crRNA (tracrRNA) are also transcribed, allowing the recognition of crRNA in the long transcript, thus directing its cleavage into CRISPR RNAs (biogenesis of crRNA). crRNAs are composed of both foreign DNA and host crRNA complexed with host tracrRNA.

In the interference phase, the crRNA/tracrRNA complex interacts with the Cas9 caspase protein which scans bacterial DNA using the crRNA as a guide. When the Cas9 finds a perfect complementarity with the viral part of crRNA, it cleaves the DNA, thus recognizing and impairing the action of the foreign sequences.

Beside the difficulties in delivering the system inside the bacteria, which can be improved by the use of cell-penetrating peptides (CPPs), nanoparticles, targeted-antibacterial-plasmids (TAPs), or phages, among others [135,137–140], the CRISPR/Cas system has some other limitations: when used to treat diseases in an organism, mutations might appear outside the target sequence causing unwanted mutations. In addition, when used to cleave target sequences, the system can act on identical or homologous DNA sequences of the host that can lead to cell death or transformation [52,141]. Its use has also been linked to the development of antimicrobial resistance [142].

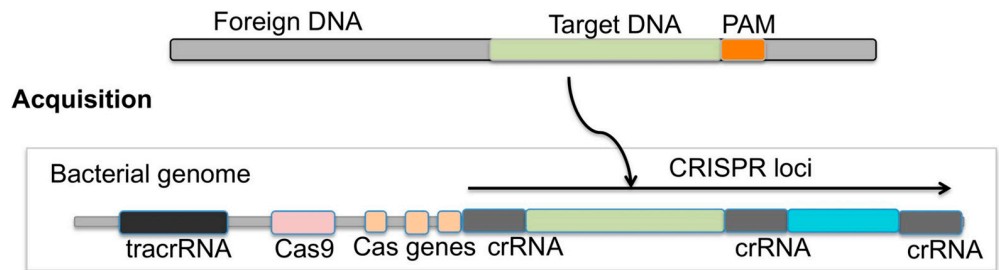

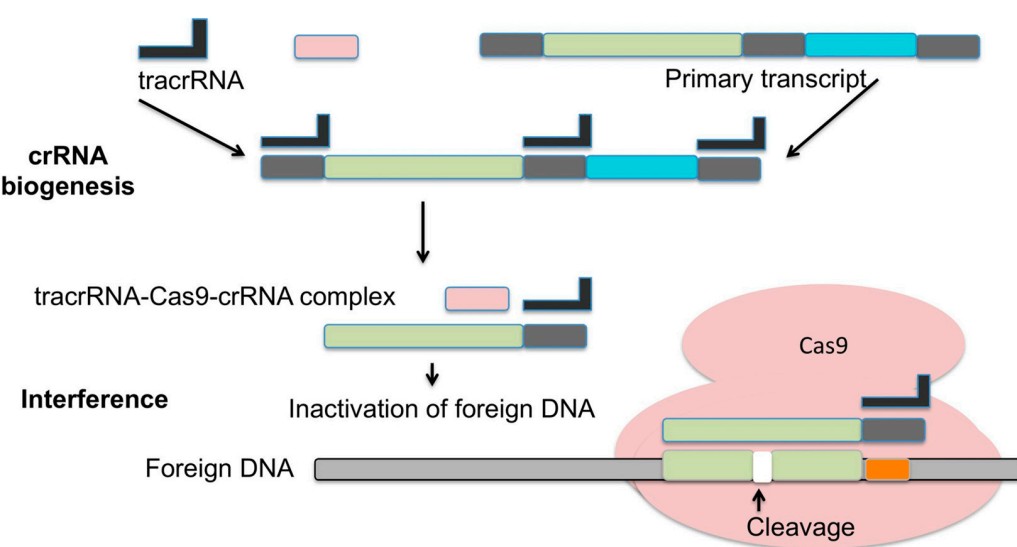

**Figure 4.** CRISPR/Cas9 mechanism of action. In the acquisition phase, fragments of foreign DNA are inserted into the CRISPR loci of the bacterial genome, next to repetitive sequences. Discrimination between self and non-self relies on the recognition of non-host sequences adjacent to the viral DNA inserted (PAM). CRISPR RNA sequences (crRNA), composed of both host and viral RNA segments, are generated by cleavage of the long transcript of the CRISPR loci [143]. The cleavage is directed by the transcription of tracrRNA, which recognizes the host portion in the long transcript based on its complementarity to crRNA. In the interference phase, the crRNA/tracrRNA system interacts with caspase Cas9, which scans the bacteria genome using crRNA as a guide in the search for foreign DNA. Once a complementarity is found, the crRNA/tracrRNA/Cas9 complex cleaves the foreign DNA (image from [143]).

2.4.5. Light-Mediated Bactericidal Techniques

The use of light of different wavelengths as an antimicrobial has been shown to eliminate a range of common pathogens, such as *S. aureus* or *Helicobacter pylori* [144]. Currently, two distinct light-mediated bactericidal techniques have been widely studied. The first is called photodynamic therapy and has shown great potential against numerous pathogens. It uses light of a specific wavelength to stimulate an exogenously supplied photosensitizer which elicits the formation of toxic reactive oxygen intermediates. The second exploits endogenous photosensitizers of the target microbe, thus eliminating the need for an external agent, but it requires a detailed knowledge of the target and its interaction with the light [144].

Antimicrobial light therapy alone or combined with a photosensitizer results in oxidative stress that leads to microbial death [144–146]. It is widely used for treating dental, skin, and soft-tissue infections. It has also been used in combination with AMPs [147,148], although additional research is needed to confirm its efficacy [10].

### 2.4.6. Silver Nanoparticles

The administration of silver nanoparticles is an intriguing antibacterial strategy, but further research is needed to assess its safety, effectiveness, and immunomodulatory effect [10]. Silver nanoparticles are able to release $Ag^+$ ions which interfere with vital processes (e.g., electron transport and signal transduction) and lead to generation of ROS, damaging bacterial constituents such as the cell wall, membrane, DNA, proteins, and biofilms, as in the case of *K. pneumoniae, P. aeruginosa, E. coli,* and *S. aureus* [149,150]. One of the main limitations of this therapy is that repeated exposure to silver nanoparticles can lead to resistance in pathogens. When used in combination with antibiotics, this phenomenon is less pronounced [10].

Other types of nanoparticles, such as those composed of zinc, platinum, or copper, have also been extensively used [151]. They can target plasmids responsible for the transference of resistance genes, sometimes in conjunction with CRISPR [137].

### 2.4.7. Microbiota Therapy

Recently, it has been reported that drug resistance can be transmitted via gut bacteria [152]. For example, *Salmonella* is able to stay in a temporary dormant state that minimizes its metabolism and prevents the action of antibiotics [153]. When conditions are favorable, the dormant cell can transfer resistance genes to other bacteria in the gut, causing a reemergence of the infection [153].

Fecal microbiota transplantation has been shown to be useful in the treatment of *Clostridium difficile* colitis [154]. However, some concerns remain since there have been reports of health complications and the growth of resistant germs [155,156].

Gut microbiota is also a source of AMPs such as bacteriocins [157], defensins [158] and others, but a healthy microbiome is required for appropriate AMP expression [159].

## 3. Viral Resistance

Because of the recent SARS-CoV-2 outbreak, the public is concerned about the development of resistant profiles in viruses. Researchers are studying resistance profiles to identify new variants [160,161], although most viruses acquire resistance to drugs and vaccines. There are many different types of pathogenic viruses which can be classified based on their genome or by the presence/absence of enveloping membranes. Viruses can contain a single strand of RNA (ssRNA) which is directly translated into proteins (RNA+ as in the case of HCV or SARS-CoV-2, although for the latter the replication mechanism is more complex [162]) or needs to be transcribed to its complementary RNA by a viral polymerase (RNA- viruses such as the influenza A, Ebola, and respiratory syncytial viruses [163,164]). Alternatively, it requires the viral enzyme reverse transcriptase (RT) to be converted into DNA and viral enzyme integrase to be integrated into the host genome, as in the case of the retrovirus HIV. Viruses can also contain double-stranded RNA (dsRNA, e.g., rotavirus), double-stranded DNA (dsDNA), as in the notable example of Herpes simplex, or a single strand of DNA (ssDNA), as in the less common parvovirus. Furthermore, their genome can be fragmented into multiple parts as in the case of influenza and HIV. The genome is protected in a capsid made of proteins, and the virus can be enveloped in a membrane which they acquire from the host. Many pathogenic viruses, such as HIV, HSV, hepatitis, and influenza, are enveloped.

In the following, we expose some examples of resistant viruses.

### 3.1. Examples of Resistant Viruses

### 3.1.1. Influenza A

Influenza A virus (IAV) is an RNA virus often treated with neuraminidase inhibitors, which prevent the detachment of viral envelopes from the cell membrane. Mutations have contributed to the development of resistance against these compounds [4,165,166], and the combination of drugs against multiple targets is currently used to bypass this problem.

Several polymerase inhibitors such as faviparavir, baloxavir marboxil, or pimodivir target each of the three subunits of the influenza RNA polymerase (polymerase basic protein 1, PB1; polymerase basic protein 2, PB2; and polymerase acidic protein, PA), which are essential for the viral replication [167]. The similarity of this protein in influenza A, B, C, and D [168] make these antivirals particularly interesting.

Baloxavir marboxil was rationally designed for binding to the PA endonuclease domain, inhibiting RNA cleavage [167,169]. One of the main issues is the development of resistances against this drug, usually caused by target mutations [167,170].

Faviparavir behaves as a mutagenic agent; it is incorporated into the nascent RNA strand as a guanosine or adenosine analog and inhibits RNA elongation. It also increases the G to A and C to U mutation rate, causing lethal mutagenesis [167]. However, K229R mutation on the PB1 subunit of RNA polymerase avoids the incorporation of favipiravir into the RNA [167].

Pimodivir acts by impairing the action of PB2, thus inhibiting RNA binding [171,172]. Some rare amino acid substitutions can impair its action, but they are rare in naturally occurring human influenza strains [167].

### 3.1.2. Hepatitis C

Hepatitis C virus (HCV) is another RNA virus possessing a high mutation rate and consequently a high genomic diversity. Moreover, in this case, the poor proofreading action of RNA polymerase results in a high mutation rate and high genomic diversity. The most common antiviral drugs for HCV inhibit either proteases or polymerases, thus affecting the viral replication [173,174].

In the last few years, multiple antiviral drugs have been developed, including protease and RNA polymerase inhibitors [175].

The first protease inhibitors were binding (covalently or reversibly) to the catalytic serine 139 in the active site of the NS3/4A RNA protease [175]. Current solutions involve macrocyclic inhibitors with better affinity and selectivity. To avoid the development of resistance, NS5A inhibitors are often used in combination with the previous. They block HCV RNA synthesis as they inhibit the formation of the replication complex in the endoplasmic reticulum and inhibit the delivery of HCV genomes to assembly sites [176,177]. Some resistant profiles are showing a lower replication rate making them hard to be detected [175].

### 3.1.3. Hepatitis B

Hepatitis B virus (HBV) is an enveloped DNA virus whose replication involves transcription to RNA intermediates that are then reverse-transcribed back to DNA. Similar to HIV and many other RNA viruses, it shows high levels of diversity in its genome, due to the lack of proofreading during reverse transcription [4,178], making it prone to develop resistance mutations.

The use of antivirals have been successful in controlling HBV replication, but the long-term use of drugs has also led to the development of resistant variants [179,180]. Most of them are focused on mutations impairing the action of the nucleoside/nucleotide analogs currently used against the polymerase. Although still effective at reducing the replication, current drugs are not curative as they do not target the nuclear DNA or suppress the HBV-infected cells. Some of the newly developed drugs are viral entry inhibitors, RNAse H inhibitors, inhibitors of nucleic acid polymers, and capsid assembly modulators [179].

Capsid assembly modulators (CAMs) are molecules able to interact with the dimers constituting the core protein (Cp) of the viral capsid. They introduce a perturbation in the assembly and reduce the production of infectious viral particles [181–183]. Two main classes of CAMs are described: CAM-A and CAM-E. CAM-A ends up with the production of aberrant structures, preventing the encapsidation of the intermediate RNA and with altered shape and size. On the other hand, CAM-E leads to the formation of chopped capsides but with a normal shape [181].

As observed for many other drugs, resistance development is promoted by mutations affecting several residues of the Cp allowing the mutant to assemble functional capsids even in the presence of the modulators [183], but work is ongoing to understand better how these changes modify the interactions between the modulator and the Cp.

### 3.1.4. Herpes Simplex

Herpes simplex virus (HSV) is a DNA virus whose genome displays low recombination rates and is replicated with high fidelity. Its infections are lifelong, cycling between periods of latency and viral shedding. Systemic antiviral therapy is usually needed by immune-compromised patients, providing a relatively successful outcome [4,184].

Acyclovir is a nucleoside analog often used against HSV, inhibiting viral DNA polymerase. Acyclovir is an acyclic guanosine analog lacking the 2′- and 3′-OH moieties. After three phosphorylation steps, the active form is obtained. Once it is incorporated into the elongating DNA chain, the chain is terminated as it lacks a 3′-hydroxyl moiety [185–187]. Different mutations in DNA polymerase confer resistance, most involving insertions/deletions [185,186]. Alternative drugs include other nucleoside analogs (as penciclovir or brivudine), pyrophosphate analogs, acyclic nucleoside phosphonates, or helicase–primase inhibitors [185,187]. As all antivirals approved for the treatment target DNA polymerase, cross-resistance has been described, making it harder to fight against resistant strains [185,186]. Several factors can increase the risk of resistance, such as long-term infections or immunosuppression [188].

### 3.1.5. Human Cytomegalovirus

Human cytomegalovirus (HCMV) is another herpes virus causing lifelong infections, often asymptomatic. As for HSV, immunocompromised patients risk the development of resistances [4]. Even though the DNA polymerase exhibits higher fidelity than RNA virus polymerases, its genome presents levels of polymorphism comparable to those of RNA viruses [189]. Mutations are often located in loci associated with envelope proteins, whose variability allows evading host immune defense. Antiviral treatment is generally based on nucleoside analogs, and resistance mainly occurs by viral-kinase phosphorylation of the pro-drug or mutations in the DNA polymerase [4,190,191].

The appearance of drug-resistant mutations in the DNA polymerase and kinase genes is problematic as it impairs the action of most of the treatments against HCMV. Most of the mutations are found in the UL54 gene encoding HCMV DNA polymerase and the UL97 gene encoding the viral kinase. Some of the most recently discovered mutations are H600L and E756G in the UL54 gene [192]. Due to these mutations, the action of current drugs (such as ganciclovir, foscarnet, maribavir, and cidofovir) is seriously impaired [190,192].

Efforts are being made to find new targets different from DNA polymerase [190,193]. The life cycle of HCMV includes the replication of the viral DNA genome, its cleavage, packaging into procapsids, and the maturation of the DNA-filled capsids. The latter are then delivered out of the nucleus into the cytoplasm by crossing the nuclear membrane. The fully assembled viral particles are finally released from the infected cells by exocytic pathways or cell lysis. These different stages are the targets for the development of novel drugs. The viral terminase complex is involved in the cleavage and packaging of the viral genome into the capsid [190]. Among the compounds targeting the viral terminase complex, benzimidazole D-riboside derivatives, phenylenediamine sulphonamide derivatives, and quinazolines are the most representative drugs [190]. Resistance mutations involving amino acid substitutions in the pUL56 subunit of the terminase complex also develop, but they are still uncommon [190].

### 3.1.6. Human Immunodeficiency Virus

Human immunodeficiency virus (HIV) is a retrovirus encoding an RNA genome that replicates in the host by reverse transcription. The process generates a copy of DNA eventually becoming double-stranded and able to insert in the host genome. Once inserted

it can be transcribed back to RNA [4,194–196]. Reverse transcriptase is error prone, thus promoting diversity and frequent resistance mutation. Even if people are usually infected by only one single clone, the high mutation rate of the HIV-1 virus leads to an exponential number of virions being produced every day in untreated individuals [196,197]. Current antiviral therapies are based on a multidrug regimen, involving inhibitors of its protease, reverse transcriptase, and integrase [4].

HIV-resistant strains can be found in two groups: people on treatment that acquire drug resistance and people who become infected with HIV-resistant strains [196,197]. It has been recently reported the infection of a patient taking pre-exposure prophylaxis with a MDR HIV-1 strain, needing combined treatment with fostemsavir for his post-infection treatment.

New strategies will be needed to face the resistance of the virus [198]. Main treatments are based on the use of nucleoside reverse-transcriptase inhibitors (NRTIs) [196]. NRTI resistance is due to the reverse-transcriptase ability to avoid the binding of the NRTI, while keeping the ability to recognize the natural dNTP substrates. Another mechanism is the promotion of the hydrolytic removal of the chain-terminating NRTI, allowing the continuation of the DNA synthesis [196,197,199].

Non-nucleoside reverse-transcriptase inhibitors (NNRTIs) interact with a hydrophobic pocket in the HIV-1 reverse transcriptase. Most of the resistance mutations are located in the domain hosting this pocket and lead to a high level of cross-resistance for this class of drugs [196,197,199].

Protease inhibitors bind to the active site of HIV-1 protease and prevent the enzyme from processing the precursors needed for viral maturation [200]. Some mutations are drug-specific, while others induce cross-resistance [197].

Integrase strand transfer inhibitors (INSTIs) block the action of the integrase, avoiding the insertion of the HIV-1 genome into the host DNA. They act by binding to the catalytic metal cations inside the active site [197]. Resistance is mediated by three main mutations: Y143C/R, Q148H/R/K $\pm$ G140S/A, and N155H $\pm$ E92Q [201,202]. Cross-resistance is commonly observed for these kinds of drugs.

Finally, some drugs block the binding of the viral gp120 protein to the host CCR5 co-receptor [199]. In this case, resistance mutations modify the pocket that is created on the viral envelope upon interaction [203].

### 3.2. Antiviral Drugs

In order to understand the action of viral drugs, it is necessary to revise the way viruses infect eukaryotic cells (Figure 5). Viral infections are accomplished in six phases: adsorption, penetration (often by endocytosis), uncoating, assembly, budding, and release [204,205].

The absorption is often mediated by a protein (e.g., the spike protein of SARS-CoV-2 [206]) recognizing a specific cellular receptor, thus explaining why infections are cell-specific. The interaction is followed by membrane fusion (in the case of an enveloped virus) and endocytosis. The viral envelope, if present, is then dissolved, freeing viral particles in the cytosol (uncoating). Uncoating is often favored by fusion of the virus-containing endosome with cellular lysosomes with low internal pH [207]. Once uncoated, RNA+ viruses can be directly used for protein expression in the ribosome machinery of the host, while RNA- viruses generally enter the nucleus and use their RNA polymerase to generate the complementary RNA+ filament (which is then used as such to reconstitute the virus but also to be translated in viral proteins by the host machinery) [163,164,205]. Retroviruses such as HIV use reverse transcriptase to produce double-stranded DNA which is then inserted into the host genome by the viral enzyme integrase [205]. DNA viruses such as HSV use their DNA polymerase to produce multiple copies of their DNA, while the host RNA polymerase transcribes the viral DNA into mRNA which is subsequently translated to viral proteins. All viral proteins and copies of the viral genome are then reconstituted in a virion (assembly), which then fuses with the host membrane (budding) and leaves the host to infect other cells (release) [205]. Notably, in the case of HIV, correct assembly

depends on the action of the viral protease, which separates the viral polypeptide into its functional proteins [208].

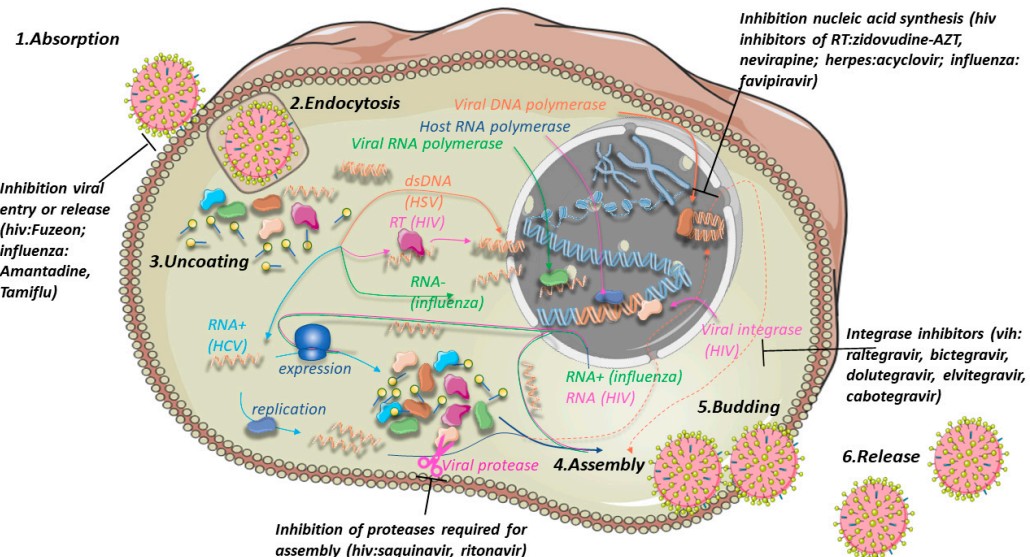

**Figure 5.** Phases of the life cycle of the different viruses (adsorption, endocytosis, uncoating, assembly, budding, and release) and a potential treatment targeting multiple processes. The mechanisms of infection of 4 different types of viruses are highlighted with arrows and related enzymes in different colors: dsDNA viruses such as HSV in orange, ssRNA retroviruses such as HIV in magenta, RNA-viruses such as influenza A in green, and RNA+ viruses such as HCV in cyan. Viral DNA is displayed in light orange, while host DNA and replicative host machinery is in dark blue. HSV assembles in the nucleus of the host (dashed lines) while its DNA can either circularize, as in the figure, or integrate into the host genome, as in the case of HIV. Own work including icons by Servier Medical Art (linear chromatin icon by Margot Riggi).

Antiviral drugs can be classified according to the phase they target. Drugs such as Tamiflu or amantadine prevent the flu virus from entering or leaving the cell [209]. Another example is Fuzeon, which blocks HIV entry [210]. Viral infections can also be treated by inhibiting nucleic acid synthesis. There exists inhibitors of HIV reverse transcriptase [211] (zidovudine-AZT, nevirapine), viral DNA polymerase (e.g., acyclovir for HSV [212]), and viral RNA polymerase (e.g., favipiravir for influenza [213]). For HIV, inhibitors of the viral enzyme integrase have been developed which prevent insertion of viral DNA into the host genome (raltegravir, bictegravir, dolutegravir, elvitegravir, cabotegravir [214]). Finally, the assembly step can be inhibited by the inhibition of its viral protease (saquinavir, ritonavir [215]).

### 3.3. Strategies to Overcome Drug Resistance in Viruses

The genetic barrier to resistance is defined as the number and type of substitutions needed to confer resistance, and it depends on the antiviral target. Consequently, a common goal in the development of new drugs (or new combinations of them) is to increase the genetic barrier by choosing targets unable to develop resistance when few modifications appear [4]. In response to new drugs, compensatory mutations are developed over time, effectively changing the viral genome. This explains why sometimes "old" drugs that had previously become ineffective can be reused. Compensatory mutations consist in restoring the altered function by new mutations or even by reversion to the original sequence [4].

An alternative approach to the use of inhibitory drugs is the development of drugs enhancing mutational processes. The strength of many RNA viruses is in their polymerase, whose fidelity of transcription is modulated to maintain their structure and allow ad-hoc modifications in response to antivirals. Some drugs increase the base mutation rate to unbalance such equilibrium, resulting in effective antiviral therapy. Ribavirin is one

example. It increases the mutation frequency as it base-pairs equally with cytosines or uridines [216,217]. Another example is molnupiravir, which has shown promising activities against different coronaviruses [218]. However, there are examples in which the fidelity of the polymerase is restored by mutations, bypassing the antiviral activity [4,219].

Some AMPs can act as antivirals by inhibiting viral enzymes, interfering with the viral assembly process or directly interacting with virions. Due to the intracellular location of viruses, CPPs, a special class of AMPs able to bypass the biological membranes, are promising agents [134,220,221].

## 4. Resistance in Eukaryotic Microorganisms

The control over the proliferation of eukaryotic microorganisms, such as parasites and fungi, is not less challenging than that of bacteria and viruses [5]. They are more similar to their hosts than prokaryotic pathogens in terms of metabolism, genetic composition, cell architecture, and biology. Additionally, many eukaryotic microbial pathogens have evolved a parasitic lifestyle which differs from the opportunistic infections characteristic of most bacterial pathogens (and fungi). This parasitism is often accompanied by the development of complex immune evasion mechanisms, which can only be faced by the action of specific immunoglobulins (IgE) [222]. Due to this capacity to evade both naturally acquired and vaccine-induced immunity, vaccines against parasites are still not effective [223]. The use of cytostatic medications is also problematic since the parasite can recover in their absence and even display immunosuppressive effects favoring the selection of drug-resistant species [5]. As in the case of several bacterial pathogens, the intracellular location of parasites makes the treatment of their infections particularly challenging [5].

### 4.1. Fungal Resistance Mechanisms

Fungal pathogens have become increasingly resistant to common antifungal compounds. Unlike antibacterial drugs, antifungal agents display restricted variability in terms of their mechanism of action, limiting therapeutic options. The development of selective non-toxic antifungal compounds is much more difficult than that of antibacterial agents, due to the similarity of fungal and animal proteins or cell structures [224]. Common antifungals are derivatives of azoles, polyenes, and echinocandins. They target ergosterol (or its synthesis) in the fungal membrane or act on cell-wall synthesis. The limited availability of antifungal drugs is problematic as pathogens become resistant to multiple classes of compounds [224–226]. Resistance mechanisms involve the expression of ATP-binding cassette (ABC) transporters [225,227] effectively reducing drug concentration in the cytoplasm, and the modification or increased synthesis of the target (often ergosterol [225,227]). Some fungi, such as *Candida*, are able to generate biofilms. Those fungal biofilms colonize abiotic surfaces and cause infections of implanted medical devices such as catheters, pacemakers, or lenses [227]. Biofilms inhibit drug penetration, and their hypoxic environment favors resistance [228].

AMPs are valid alternatives to common antifungals as they can disrupt fungal cell membranes or even act intracellularly [229,230] by targeting fungal mitochondria and induce apoptosis [224].

### 4.2. Drug Resistance in Parasites

Parasitic diseases have an enormous impact on human health. Some of the most challenging are malaria (*Plasmodium*), leishmaniasis, sleeping sickness (*Trypanosoma brucei*), Chagas disease (*Trypanosoma cruzi*), and toxoplasmosis (*Toxoplasma*), among others [231,232]. The benefits of available drugs are being threatened by the development of parasite drug resistance [231]. Targeting their mitochondria with CPPs, a subclass of AMPs, might constitute a valuable alternative to common drugs.

In the case of malaria, most of the drugs act against the intraerythrocytic development of *Plasmodium*. It is critically necessary to create medications that stop the transmission and suppress their asymptomatic and hepatic versions [5,231]. Resistance to antimalarial

drugs is documented in three of the five malaria species affecting humans. For example, resistance to chloroquine, the "gold standard" for malaria, consists in the activation of efflux pumps preventing its internalization [5,231]. Cross-resistance between drugs of the same chemical family or sharing similar modes of action has been observed [5,231].

Another issue with several antiparasitic medications is their high toxicity. That is the case of many leishmaniasis treatments displaying pancreatic, cardiac, and renal toxicity [231]. Cellular resistance mechanisms include decreased drug uptake or increased efflux, drug inactivation, and target gene amplification [5,231]. Interestingly, some Leishmania resistant strains may lower their ergosterol content and substitute it with stigmasterol [233].

Trypanosomes have an amazing ability to adapt to drug pressure. They are able to lower drug levels inside their cells, alter the molecular target of the drug, and develop general defense and repair mechanisms [5,231]. The treatment of trypanosomal infections depends on the stage of the disease. While safe drugs are preferred in the first stage of infection, the second stage requires drugs capable of crossing the blood–brain barrier to reach the parasite in case of brain infections, and they are generally toxic and difficult to administer [5,231,232].

*Giardia lamblia* is one of the most common and sometimes challenging intestinal parasites of humans and animals [234]. Therapeutic failures have been observed with all the common anti-*Giardia* agents, due to reinfection, inadequate drug dose, immunosuppression, and drug resistance [234]. Such issues have been reported worldwide but especially in Asia or in the Mediterranean area [234].

Common resistance mechanisms include drug elimination and deactivation [234]. Some AMPs, such as indolicidin, have been proved useful against this parasite [235].

### 4.3. Cancer-Cell Resistance

The acquisition of endogenous multidrug resistance (MDR) by cancer cells (Figure 6) is a common problem in anticancer therapies. Numerous resistance mechanisms exist, including resistance to radiotherapy [236].

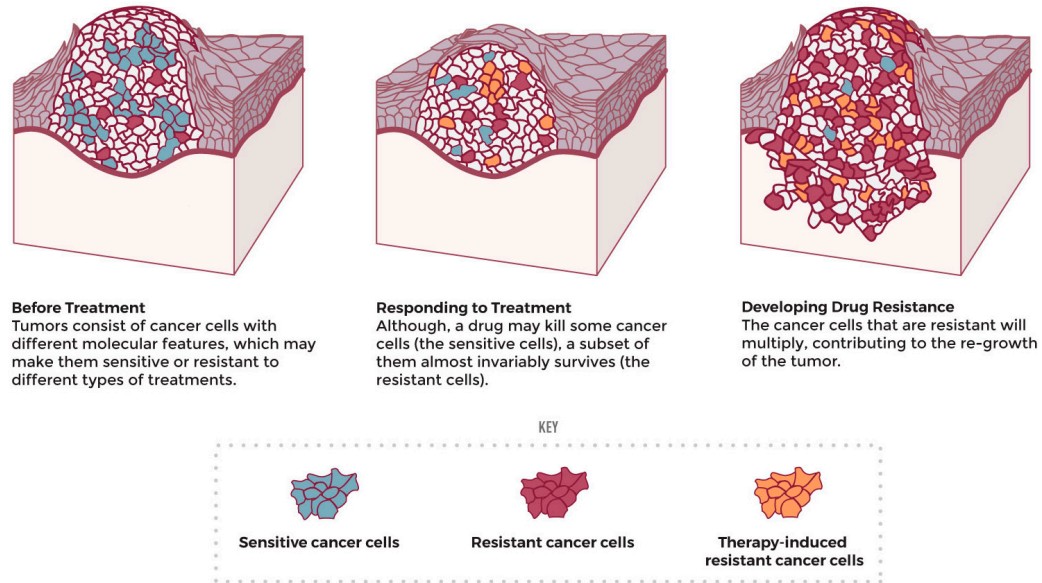

**Before Treatment**
Tumors consist of cancer cells with different molecular features, which may make them sensitive or resistant to different types of treatments.

**Responding to Treatment**
Although, a drug may kill some cancer cells (the sensitive cells), a subset of them almost invariably survives (the resistant cells).

**Developing Drug Resistance**
The cancer cells that are resistant will multiply, contributing to the re-growth of the tumor.

KEY

Sensitive cancer cells      Resistant cancer cells      Therapy-induced resistant cancer cells

**Figure 6.** Drug resistance development upon treatment (adapted from the National Cancer Institute [237]).

Chemotherapy is associated with numerous severe side effects, which include immediate signs of toxicity and late signs of chronic toxicity. According to the WHO classification, the intensity of the signs of toxicity can be mild (grade 1), moderate (grade 2), severe (grade 3), and life-threatening or disabling (grade 4). Grade 3 and 4 neurotoxicity can induce somnolence, paralysis, ataxia, and spasms, among other effects. In addition, the chronic effects

of chemotherapy include drug resistance, carcinogenicity, and infertility [238]. Common chemotherapy treatments involve the use of fluoropyrimidines, cisplatin or derivatives, and taxanes. Due to their important toxicities [239], it is essential to create new, more specific agents.

We will revise here the main classes of anticancer agents (Figure 7) [240] and discuss the resistance phenomena associated.

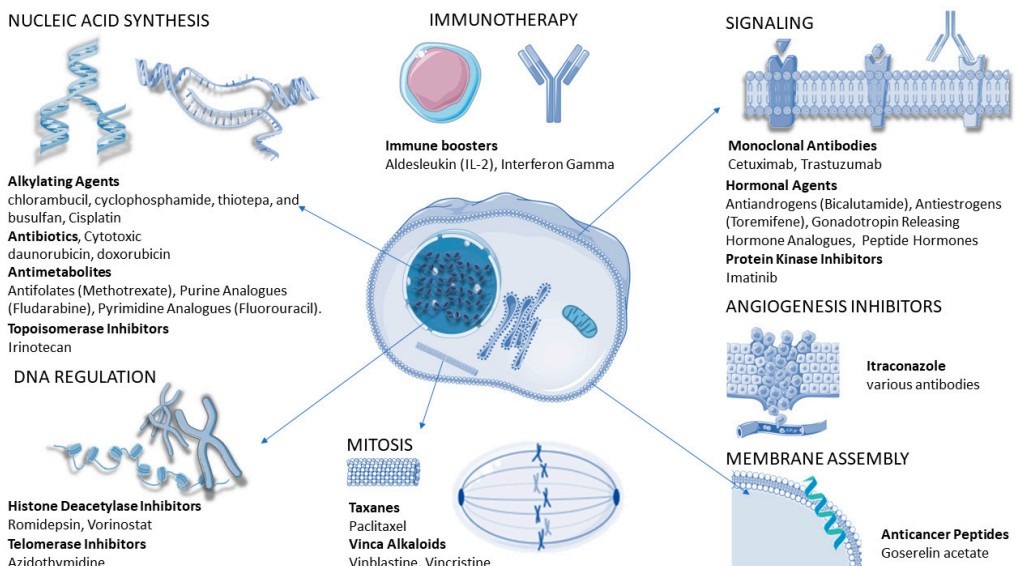

**Figure 7.** Different classes of anticancer drugs and their modes of action. Own work including icons by Servier Medical Art (linear chromatin by Margot Riggi, alpha helix by DBCLS).

Cytostatic drugs are commonly used to target the cell cycle. They can be grouped into alkylating agents which damage DNA (e.g., cisplatin), antimetabolites that replace the normal building blocks of RNA and DNA (e.g., capecitabine or 5-fluorouracil), inhibitors of DNA replication enzymes, inhibitors of enzymes involved in unwinding DNA during replication and transcription (topoisomerase I or II), inhibitors of mitosis and cell division (e.g., paclitaxel or docetaxel) [241], and corticosteroids for the relief from side effects caused by chemotherapy [238,242]. Furthermore, many antibiotics are able to intercalate in DNA, inhibiting its replication or translation [243].

In particular, the alkylating agent cisplatin is a platinum-based compound that forms intra- and inter-strand links with DNA, inducing cell-cycle arrest and apoptosis in most cancer cell types. While effective in many cases, drug resistance can be developed [242].

Taxanes such as paclitaxel and docetaxel are able to perturb the dynamic structures of microtubules, constantly incorporating and releasing soluble dimers of tubulin [241]. They also face the development of resistance [242]. Resistance to drugs such as paclitaxel and vincristine (blocking mitosis by inhibition of tubulin disassembly or assembly, respectively) or doxorubicin (which intercalates DNA and blocks protein synthesis) can be caused by the ABC transporter family, which enhances their outflow and reduces their intracellular accumulation [6,244,245].

Drugs inhibiting the histone deacetylase promote hyperacetylation of the histone, which in turn induces apoptosis [246,247]. On the contrary, inhibitors of telomerase (an enzyme which adds guanine-rich repetitive sequences to telomeres and prevents chromosomal instability) act on cancer cell replicative capabilities [248–251].

Other drugs interfere with cellular signaling. Hormone-mimicking agents compete with hormones, inhibiting their cell-division-promoting action. Similarly, monoclonal antibodies are able to bind to cell receptors, blocking the action of growth factors or inhibiting signals leading to angiogenesis [252]. Finally, kinase inhibitors block the intracellular signal cascade leading to the gene expression for cell division [253].

In the search for new active compounds, anticancer peptides (ACPs, which can be considered a particular class of AMPs) are promising candidates, due to their low toxicity, selectivity for cancer cells, and lower tendency to induce resistance compared to chemically synthesized agents [244,254,255]. This is due to their peculiar complex target (the cell membrane) whose modification normally requires complex reorganization. ACP peptides often recognize specific phospholipids on cancer cell membranes [21], such as PS, PE, and PI [256–262]. In other cases, their action is exerted by interaction with specific proteins; they can bind to growth-factor receptors inhibiting growth, block the action of transporters, promote antiangiogenic effects, and inhibit enzymes (such as kinases or proteases) which regulate the growth, invasion, and metastasis of cancer cells [244,254,255]. Finally, ACPs can be rapidly cleared from the blood and non-targeted tissues [255], and their immunomodulatory properties can enhance immunotherapies [263,264], favoring the action of immune cells [254,265].

Immunotherapy has gained significant attention due to its effectiveness on several recalcitrant cancer types, and it is used as a last-resort strategy when other approaches fail. On the other hand, immunotherapy can cause important undesirable autoimmune effects on healthy tissues [254,266]. Resistance to immunotherapy involves the evasion of immune recognition and the development of cancer immunoediting processes [254,266,267], leading to the selection of resistance clones during the progression of tumors. The strength of the patient is also important because evasion can occur more easily in immunosuppressed patients [267]. Some of the mechanisms of resistance that tumors can develop are the absence of antigen presentation, underexpression of antigenic proteins, and insensibility to T cells, among others [266].

Novel approaches are based on interference RNA (RNAi), which has shown promising results against esophageal adenocarcinoma [268] and others [269]. Gene silencing is a natural process that can be used to inhibit gene expression in a sequence-specific manner [268]. Silencing can be achieved by RNAi based on the action of microRNA (miRNA) or small interfering RNA (siRNA), short duplex RNA molecules able to degrade mRNA or inhibit its translation with a variable degree of specificity (at variance with miRNA, siRNA are highly specific). miRNAs are noncoding single-stranded RNAs that regulate the expression of complementary messenger RNAs and play a role in physiological and pathological pathways. They can control cell proliferation, apoptosis, metastasis, and angiogenesis, and can be used to prevent the activation of pro-survival pathways activated by resistant cancer cells [270]. siRNAs are short antisense RNA molecules that inhibit the expression of target genes by inducing the degradation of mRNA molecules. siRNA can be introduced into cells by direct administration or plasmid/viral vector systems [268]. Viral vectors are efficacious delivery systems although they are potentially tumorigenic and immunogenic, besides having limited cargo-carrying capacities [259,271]. AMPs have shown their viability in the fight against cancer, and they are also valuable peptide alternatives as delivery systems [259–262,271,272].

Other resistance mechanisms aim at lowering the expression of transporters granting intracellular access to the drug or reducing its binding by mutation of the target [6,245]. Tumoral cells can also upregulate the anti-apoptotic genes (e.g., Bcl2, AKT), downregulate the pro-apoptotic genes (e.g., Bax, Bcl-xL), and activate DNA repair systems (excision repair system and homologous recombination repair mechanisms) when treated with cisplatin or amplify genes targeted by anticancer agents. The overexpression of the high-mobility group A (HMGA), a subgroup of non-histone chromatin-associated proteins, is also linked with metastases and the development of chemoresistance [273,274]. Furthermore, resistance can rely on the methylation of DNA or histone alterations (epigenetics) rather than mutations of target proteins [6,245].

Mitochondria in Cancer

Apoptosis, whose inhibition is one resistance mechanism, allows the stop of the uncontrolled proliferation of non-functional cancer cells. Triggering mitochondria-induced apoptosis by means of drugs is therefore a promising anticancer strategy [66,275,276].

Mitochondria have their own double-stranded circular DNA (mtDNA) genome which can possibly mutate due to their exposition to ROS and the absence of mitochondrial histones [277]. The deregulation of enzymes caused by such mutations leads to the accumulation of intermediary metabolites accelerating tumorigenesis [66,276]. Mutations can also impair the activity of YB-1 in human mitochondria, an enzyme repairing DNA mismatches, resulting in mtDNA microsatellite instability (mtMSI) which in turn causes frequent frameshift or missense mutations in cancers such as colorectal carcinomas. Most of the mtMSI results in truncated proteins that affect mitochondrial metabolism and favor carcinogenesis [276]. Mutations in DNA polymerase-gamma, the enzyme that synthesizes mtDNA, are found in almost all cancers, causing depletion and mutations in mtDNA. Other consequences include an alteration in the mitochondrial membrane potential and an increase in ROS generation [276].

Mitochondria in cancer cells are structurally and functionally different from their counterparts in healthy cells, making them an interesting target for new drugs. Some take advantage of the high membrane potential and low expression of $K^+$ channel in cancer cells; others target oxidative phosphorylation complexes. Another strategy is targeting mitochondrial ribosomes (which are similar to bacterial ones) by antibiotics interacting either with the large or the small subunit of bacterial ribosomes [30,66,278].

There exists multiple ways to deliver drugs to mitochondria. Nanoparticles of gold N-heterocyclic carbene complexes are able to target both cancer cells and mitochondria [279]. Alternatively, a special subclass of peptides (mitochondria-penetrating peptides, MPPs) can deliver to their interior drugs [255] which can subsequently be released photothermally [280,281] or activated by photosensitizers for radiotherapy [282]. Active compounds can either damage mtDNA for anticancer activity [66] or even be transported to mitochondria to prevent their degradation by the cytoplasmic enzymes [283].

## 5. Novel Drug Delivery Systems

Every day, new delivery mechanisms are being created in response to the issue of resistance. While it is hard to explain all of them, some play an important part in the battle against not only bacteria, but viruses, fungi, and parasites as well. They involve the use of lipidic systems, various types of nanoparticles, dendrimers, AMPs, and so on. All of these delivery methods aim to preserve the optimal drug concentration at the target for a sufficient amount of time [284].

Polymeric nanoparticles are often manufactured from polymers such as polyethylene glycol, polylactic acid, or polycyanoacrylate; however, polysaccharides such as cyclodextrins or chitosan are gaining popularity [285–287]. They have the benefit of not being identified by the immune system and hence not being destroyed by phagocytosis. Some nanoparticles are composed of phospholipids, which allows delivery of lipophilic drugs, enhancing their pharmacokinetics [284,288]. Inorganic nanoparticles are made from nanomaterials such as gold, silica, and carbon nanotubes, among others. They are chemically inert, making them non-toxic. Some display magnetic properties that may be exploited to direct them to the target, thus improving medication release [79].

Liposomes may also be utilized as delivery systems; they are a lipid layer with an aqueous center that can be used to convey medications based on their affinity for the hydrophilic core or the lipid layer. They have been used to carry anticancer drugs, or for mRNA vaccines [284,289,290], but they can also enclose antimicrobial molecules against various microorganisms [284,291–293]. Their limited stability can be partially addressed by coating them with polymers or modifying their charge [284,291,292]. Other lipidic systems include micelles (that can carry lipophilic drugs [294]) or cell-produced or synthetic [295] exosomes. Many cells naturally produce exosomes, which are endocytic vesicles able to fuse

with cell receptors and release their contents. They have been used to deliver drugs against pathogens and cancer cells, or for the treatment of neurodegenerative disorders [296].

Nanosuspensions are colloidal systems often used to deliver lipophilic drugs. They are prepared by media milling under a high-pressure homogenization [297]. Nanoemulsions are mixtures stabilized by the addition of emulsifiers that can be used for the treatment of respiratory infections caused by challenging bacteria such as *K. pneumoniae*, *A. baumannii*, *S. aureus*, or *Mycobacterium tuberculosis* [284,298]. Pulmonary surfactants, composed of lipids and four specific lung surfactant proteins, have been used to reach distal airways. They can spread quickly and efficiently deliver antivirals [299], AMPs [300,301], or anticancer drugs [302] at the air–liquid interfaces.

Dendrimers are non-cytotoxic branched molecules containing a hydrophobic core from which repeating units form dendritic branches [284]. They are able to deliver hydrophobic drugs in the core cavity and hydrophilic drugs on the large surface formed by the chains [79]. Similarly, nanofibers are polymeric biomaterials produced by different methods (laser spinning, self-assembly, electrospinning, etc.) with a large surface of contact that can carry multiple active molecules (antibiotic, anticancer drugs, AMPs [303]) for oral and transdermal drug delivery [304].

A recent approach is based on the use of empty bacterial envelopes from Gram-negative bacteria, often named "bacterial ghosts". Because of the presence of pathogen-associated molecular patterns (such as LPS, lipoprotein, peptidoglycans), they can be used for the delivery of nucleic acids, drugs, and antigens to immune cells. Not only do they protect the entrapped cargoes but also train immune systems against infections and tumors [305].

Microcapsules are gaining attention due to their high loading capacities, semi-permeable character, and biocompatibility, but the leakage of drugs, which is based on changes in pH and temperature at the target site, is still a major challenge. Ultrasounds can be used to facilitate their tissue penetration depth and spatiotemporal control [306,307].

Molecularly imprinted polymers (MIPs) contain recognition sites which bind target molecules [308]. They behave as "synthetic antibodies" as they bind to the antigen with high affinity and selectivity. They are created by the co-polymerization of functional and cross-linking monomers with a template molecule [308]. The functional monomers assemble around the template, followed by polymerization with a cross-linker. The template is finally taken out of the 3D polymer, revealing the cavities that were imprinted [308]. Cavities complement the template in terms of the size, shape, and positioning of the chemical groups [309]. One of the major drawbacks in protein imprinting is the modification of the protein's native conformation and the challenges in the generation of MIPs for certain sites of the protein [309]. Epitopes, which are protein fragments that are often exposed at the surface, have been successfully exploited to create MIPs that recognize and bind to the complete protein effectively [309]. MIP binding at specific sites can be obtained by using short peptides reproducing unstructured protein sequences, as in the case of epitopes of the hepatitis A virus cell receptor-1 [309]. The same strategy has been applied to inhibit cell–cell adhesions [310], demonstrating potential applications of MIPs as antimicrobial or antitumor agents [311,312].

## 6. Conclusions

In the present work, we have explored the challenges of resistance found in many different microorganisms (virus, bacteria, fungi, parasites), covering also resistance in cancer cells. Throughout this large spectrum of living systems, many common mechanisms are at work: drug degradation, target modification, the expression of systems able to eject drugs out of the cell, or the reduction in cell permeability and/or accessibility. In the case of cancer cells, additional mechanisms have to be considered, which include prevention of apoptosis, overexpression of oncogenes, underexpression of tumor-suppressor genes, insensitivity to signals limiting growth, replicative immortality, and hijacking of the immune system.

Due to the different architecture of bacteria as compared to animal cells, antibacterial drugs can act on multiple targets without causing severe toxicity. On the contrary, drugs targeting eukaryotic pathogens and cancer cells are often toxic, and new strategies have to be developed to limit side effects.

In all cases, many AMPs have proven a valuable alternative to conventional drugs in that they selectively target cancer cells, enveloped viruses, fungi, or bacteria while leaving healthy animal cells unaffected. Part of the reason for their selectivity stems from the differences in the superficial charge of their targets (neutral in most animal cells and negative in most microorganisms). Other factors are crucial, including the amino-acid sequence of the AMP, the fluidity of the membrane, or the specific recognition of a molecular target (ergosterol, intracellular protein, etc.). Last but not least, the development of resistance is unfavored by the heterogeneity of their membrane target. In this scenario, the creation of new tools to potentiate their pharmacological properties (cost of production, bioavailability, biostability) are urgently needed, as reflected by the huge efforts in the development of novel delivery systems.

**Author Contributions:** N.D.: investigation, writing—review and editing, visualization, project administration, funding acquisition; F.R.-M.: investigation, writing—original draft, writing—review and editing, visualization. All authors have read and agreed to the published version of the manuscript.

**Funding:** Francisco Ramos-Martín's PhD fellowship was co-funded by Conseil régional des Hauts-de-France and by the European Fund for Economic and Regional Development (ERDF).

**Institutional Review Board Statement:** Not applicable.

**Informed Consent Statement:** Not applicable.

**Data Availability Statement:** Not applicable.

**Conflicts of Interest:** The authors declare no conflict of interest.

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
