# Peer review of "Drug Resistance: An Incessant Fight against Evolutionary Strategies of Survival"

_2036-7481, doi:10.3390/microbiolres14020037_

Round 1

Reviewer 1 Report

Well written and informative manuscript in the field of microbial resistances. Please add a section of novel drug delivery technologies to overcome microbial resistance. Please add a section of novel drug delivery technologies to overcome microbial resistance.

Author Response

Please find the answer in the attached file.

Reviewer 2 Report

The review article entitled "Drug resistance: an incessant fight against evolutionary strategies of survival" is indeed a well written and informative review. 

I am only concerned about the way the authors describe different resistant bacteria and viruses. For instance,  in section 2.2, the antibiotic resistant strains of bacteria has been described. This can be made into a table format. If writing as paragraph, the features and resistance mechanisms of each strains should be elaborated. 

Similarly, in section 3.1, each viruses described should be elaborated with its potential resistance mechanisms, or the strains should be described as a table.

In section 2.4, authors describe different strategies to overcome bacterial resistance to antibiotics. Each section in this should be elaborated with at least one example described from the references. This is important as the manuscript deals with review about drug resistance

Section  2.4.2, The authors have mentioned certain disadvantages of phage therapy. Why not to add a separate section detailing the use of Phage derived enzymes or endolysins or enzybiotics against drug resistant bacteria.  The authors could also add this in in section 2.4.6, where they describe antimicrobial peptides therapy.

Author Response

(The authors gave the same response as above.)
